# Regionally Compatible Individual Tree Growth Model under the Combined Influence of Environment and Competition

**DOI:** 10.3390/plants12142697

**Published:** 2023-07-19

**Authors:** Wenjie Zhang, Baoguo Wu, Yi Ren, Guijun Yang

**Affiliations:** 1School of Information Science and Technology, Beijing Forestry University, Beijing 100083, China; zwenjie@bjfu.edu.cn; 2Key Laboratory of Quantitative Remote Sensing in Agriculture of Ministry of Agriculture and Rural Affairs, Information Technology Research Center, Beijing Academy of Agriculture and Forestry Sciences, Beijing 100097, China; 3National Engineering Research Center for Information Technology in Agriculture, Beijing 100097, China; 4Forestry Information Research Institute, Beijing Forestry University, Beijing 100083, China; 5Academy of Forestry Inventory and Planning, Beijing 100714, China; xue.qin@syngentagroup.cn

**Keywords:** regionally compatible, competition, environmental factors, individual tree growth

## Abstract

To explore the effects of competition, site, and climate on the growth of Chinese fir individual tree diameter at breast height (DBH) and tree height (H), a regionally compatible individual tree growth model under the combined influence of environment and competition was constructed. Using continuous forest inventory (CFI) sample plot data from Fujian Province between 1993 and 2018, we constructed an individual tree DBH model and an H model based on re-parameterization (RP), BP neural network (BP), and random forest (RF), which compared the accuracy of the different modeling methods. The results showed that the inclusion of competition and environmental factors could improve the prediction accuracy of the model. Among the site factors, slope position (PW) had the most significant effect, followed by elevation (HB) and slope aspect (PX). Among the climate factors, the highest contribution was made by degree-days above 18 °C (DD18), followed by mean annual precipitation (MAP) and Hargreaves reference evaporation (Eref). The comparison results of the three modeling methods show that the RF model has the best fitting effect. The R^2^ of the individual DBH model based on RF is 0.849, RMSE is 1.691 cm, and MAE is 1.267 cm. The R^2^ of the individual H model based on RF is 0.845, RMSE is 1.267 m, and MAE is 1.153 m. The model constructed in this study has the advantages of environmental sensitivity, statistical reliability, and prediction efficiency. The results can provide theoretical support for management decision-making and harvest prediction of mixed uneven-aged forest.

## 1. Introduction

To obtain information on the potential future productivity of plantation forests, forest managers rely on various types of growth models [1]. The stand level growth model can be used to predict the development of even-aged forests, but for mixed forests or uneven-aged forests the individual tree growth model is required [2]. The individual tree growth model is a bottom-up modeling method, starting from individual trees in the system and ending at the stand level, aiming to reveal and predict the growth mechanism of individual trees [3]. Compared with pure even-aged forests, the complexity, depth, and breadth of the forest growth simulations for mixed uneven-aged forests increased, and the scale gradually shifted from stands to individual trees [4]. And further compared with the whole stand model, the individual tree model is more applicable [5], promoting the widespread use of individual tree growth models in forest management [6].

According to the concept of mathematical statistics, the model can be divided into parametric model, non-parametric model, and semi-parametric model. A parametric model mainly has the form of an algebraic equation, differential equation, and transfer function [7]. It is widely used in forestry because of its fast modeling speed and small amount of data required. With the development of computers, non-parametric models have expanded the application range of forestry models with their strong applicability and lax requirements on model assumptions, such as non-parametric additive model [8], the classification and regression tree (CART) [9], support vector machines (SVM) [10], random forest (RF), and neural network (NN). In recent years, semi-parametric models have been gradually applied to the study of tree mortality and inside boundary timber, such as the Cox proportional hazards model [11] and semi-parametric geographically weighted Poisson regression (SGWPR). The semi-parametric model contains a parametric part and non-parametric part, which overcomes the deficiency of the parametric model and non-parametric model and solves the “dimensional disaster” of the non-parametric model and other problems. However, due to the relative complexity of the semi-parametric model, there are many problems from theoretical research to practical application. At present, the construction of tree growth models is still dominated by parametric models and non-parametric models, and semi-parametric models are rarely studied.

Regression analysis is the most commonly used method for the modeling of individual tree growth models [12,13,14], but the traditional regression method is usually based on certain statistical assumptions, such as data independence, normal distribution, and equal variance. Forest growth data have the characteristics of continuous observation and hierarchy, which makes it difficult to satisfy the assumptions in general. As a result, it is difficult for traditional regression models to achieve higher prediction accuracy; so, it is necessary to try new modeling methods [15]. With the development of data processing technology, artificial intelligence has provided more advanced technical ideas to overcome such problems. An artificial neural network was used earlier in individual tree growth modeling [16], which can approximate any nonlinear trend to the maximum extent [17]. In particular, the BP neural network (BP) is the most commonly used artificial neural network in model research due to its strong generalization ability, high accuracy, and small error [18]. In addition, random forest (RF) in ensemble learning has also been gradually applied in forestry research [19]. RF can be highly parallelized in training and can provide the feature importance of each factor on the output. RF has a strong model generalization ability and is not sensitive to partial feature loss [20]. Weiskitte [21] studied the climate-driven site index with RF. Kilham [22] applied RF to the selection of felled trees and the prediction of stand accumulation. Currently, RF is widely applied in forestry remote sensing and other fields; however, it is rarely applied to the prediction of forest growth and harvest.

Tree growth is affected by many factors, among which genotype, climate, and site are the main driving factors [23,24]. Therefore, environmental factors should be included in the growth model to explain the growth pattern and improve the accuracy of tree growth prediction [25,26]. The individual tree growth model usually includes a competition index to quantify the degree of competition pressure of trees in the stand [27]. Generally speaking, adding competition effects to the growth model can improve the model performance [28]. Many current studies add a competition index dependent on distance or independent distance to the tree growth model to quantify competition [29]. It is used to predict tree diameter at breast height (DBH) growth or tree height (H) growth [30,31,32], which improves the accuracy of the model. Although some studies have shown that the growth of trees is obviously correlated with the distance of surrounding trees [33,34], some scholars have found that the competition index of independent distance is more effective in fitting the growth of trees [35], which may be related to the difference in research objects and the calculation of the competition index.

Site is an important environmental factor affecting tree growth, which determines the water, heat, and soil fertility and is the core factor for the growth and distribution evolution of trees under natural conditions [36,37]. For example, slope gradient leads to light heterogeneity [38]. Slope aspect can affect light, temperature, and soil physical and chemical properties and then affect tree growth and stand structure [39,40]. The influence of soil on plants is also well known; nitrogen, phosphorus, potassium and other nutrients, and humidity in soil vary with soil thickness and soil type [41]. Adding site factors into the growth model is of great significance for selecting reasonable afforestation tree species, increasing ecological stability, and supporting forest productivity [42,43]. Climate is an important external environment that determines the dynamic change of tree distribution and forest function at the regional scale [44,45]. Under the same site conditions, different climates will lead to different tree growth. Therefore, it is crucial to add climate factors into the growth model to improve its universality [46,47].

Close-to-nature forest management is a feasible theory and technology to improve the stability of forest ecosystems [48]. Building mixed uneven-aged forest is one of the main management measures of close-to-nature forest management. In order to build a mixed uneven-aged forest, it is necessary to conduct management operations on the forest, such as tending and thinning, adjusting the stand structure, etc. One of the important problems to be solved is to estimate the results of management operations, such as the harvest volume and biomass, etc. The solution is to build an individual tree growth model. There are many factors affecting forest growth, and it is very important to study the co-driving mechanism of competition and the environment to predict tree growth. However, there are much research on grassland and aquatic ecosystem, but there is little research on forest ecosystems [49,50] and most of them focus on the unilateral effects of stand density, climate factors, and tree species composition on tree growth. There are few individual tree growth models that include competition, site, and climate factors.

Chinese fir (*Cunninghamia Lanceolata (Lamb.) Hook.*) is one of the most important timber species in the south of China. The results of the ninth national continuous forest inventory (CFI) show that the area of Chinese fir plantations accounts for 1/4 of the total area of China’s artificial arbor forest, and the planted area ranks the first among afforestation tree species [51]. As one of the six large forest regions in China, Fujian Province is located in the subtropical climate zone along the southeast coast of China, with complex and changeable climatic conditions and frequent extreme weather and climate events, making it a highly sensitive area to climate change [52]. Therefore, we took Chinese fir as the study tree species and Fujian Province as the study area to reveal the mechanism of competition factors (Comp), site factors (Site), and climate factors (Clim) effects on individual trees. The aims of this research study are: (1) to quantify the influence degree of factors, in order to select the relatively important factors in the individual tree growth model; (2) to explore the use of the re-parameterized (RP) method, BP, and RF algorithm to construct an individual tree growth model, which compares the accuracy and adaptability of different methods; and (3) to add the screened factors into the model and construct a regionally compatible individual tree growth model under the combined influence of environment and competition. The results of this study can provide references for the construction of individual tree growth models for different tree species in different regions, provide model support for growth harvest prediction of mixed uneven-aged forests, and also have indirect support significance for predicting forest carbon stocks and rational response to climate change.

## 2. Results

### 2.1. Individual Diameter Growth Model Based on RP

With DBH as the dependent variable and T as the independent variable, the Gompertz model, Logistic model, Mitscherlich model, and Richards model were selected and fitted using the least squares method, respectively. SPSS software [53] was used to fit the data, and the R^2^ and RMSE were used to evaluate the fitting results. As can be seen from the Appendix A, all of the above four basic models except Richards could converge, and all model parameters could pass the significance test. The Mitscherlich model had the best fitting effect, with the largest R^2^ and the smallest RMSE.

Therefore, the Mitscherlich model was selected as the optimal basic model, and Comp, Site, and Clim were added to construct the individual diameter growth model based on RP (DBH-RP model). Its parameter expression is shown in Equation (1).
(1)$$y=\omega_{0}*(1-e^{-\omega_{1}*T})$$

On the basis of the previous analysis, R software was adopted to construct a stepwise regression using the step function. The re-preference of the independent variables was completed based on the Akaike information criterion (AIC). The final model independent variables obtained include sample plot tree density (N), slope position (PW), elevation (HB), soil thickness (TRHD), landform (DM), degree-days above 18 °C (DD18), and Hargreaves reference evaporation (Eref). Therefore, parameters ω0 and ω1 could be expressed as a linear combination of the above variables. The results of parameter estimation of the DBH-RP model estimation are shown in Table 1.
(2)ω0=α0+α1∗N+α2∗PW+α3∗HB+α4∗TRHD+α5∗DM+α6∗DD18+α7∗Eref
(3)ω1=β0+β1∗N+β2∗PW+β3∗HB+β4∗TRHD+β5∗DM+β6∗DD18+β7∗Eref

After processing, the DBH-RP model including T, Comp, Site, and Clim with statistically significant parameters was as shown in Equation (4).
(4)$$D=\omega_{0}*(1-e^{-\omega_{1}*T})$$

In the formula, ω_0_ and ω_1_ were parameters. The formula is shown in Equations (2) and (3). The values of the coefficients in the formula are shown in Table 1.

In the test data set, R^2^ of the DBH-RP model was 0.621 and RMSE was 2.813 cm; R^2^ was 12.3% higher than that of the basic model, and RMSE was 9.3% lower. The predicted change in DBH with T of the DBH-RP model is shown in Figure 1.

The residual analysis was used to evaluate the model, and statistical diagnosis was made on the model through the state of the residual graph. The residual comparison between the Mitscherlich model and the DBH-RP model is shown in Figure 2.

As can be seen from Figure 2, the residual was evenly distributed between −6.0 cm and 6.0 cm, but after the addition of Comp, Site, and Clim and re-parameterization, more residuals were reduced to between −4.0 cm and 4.0 cm on the 5–20 cm DBH interval, but on the DBH interval beyond greater than 20 cm, residuals did not change significantly. There was no heteroscedasticity in either the Mitscherlich model or the RP model.

### 2.2. Individual Diameter Growth Model Based on BP

The individual diameter growth model based on BP (DBH-BP) was trained. Table 2 shows the fitting results of the sub-model and the whole model based on BP. 

When the independent variable only includes T, the model can explain 63.8% of the DBH variable, and it is concluded that T was the most important factor affecting the growth of DBH. However, after Comp, Site, and Clim were introduced into the model, the predictive ability of the model improved. R^2^ increased by 7.9%, 12.6%, and 4.1%, respectively, while MAE and RMSE decreased. The predictive accuracy of the whole model composed of all factors reached the maximum, which could explain more than 80.8% of the DBH variable. These results indicate that the introduction of Comp, Site, and Clim could improve the prediction accuracy of the individual tree growth model. The predicted change in DBH with T of the DBH-BP model is shown in Appendix A.

Meanwhile, it can be seen from the distribution diagram of the residuals of the four models (Figure 3 and Appendix A) that they were evenly distributed without obvious heterogeneity. However, with the addition of Comp, Site, and Clim to the model, the residual error of the model was gradually reduced from −6.0 cm and 6.0 cm to −3.0 cm and 3.0 cm, and the model accuracy was improved.

### 2.3. Individual Diameter Growth Model Based on RF

RF model was used to build an individual diameter growth model (DBH-RF). Table 3 shows the sub-model and whole model fitting results using RF model.

Table 3 shows that when the independent variable contained only T, the model could explain 67.9% of the DBH. However, after Comp, Site, and Clim were introduced into the model, the predictive ability was improved, R^2^ was increased by 8.0%, 12.7%, and 2.8%, respectively, and MAE and RMSE were decreased. The prediction accuracy of the whole model composed of all factors reached the maximum, which could explain more than 84.9% of the DBH variable, and MAE and RMSE were the minimum. The predicted change in DBH with T of the DBH-RF model is shown in Appendix A.

At the same time, it can be seen from the distribution diagram of the prediction and residuals of the four models (Figure 4 and Appendix A) that the residuals of the four models were evenly distributed without obvious heterogeneity. However, with the addition of Comp, Site, and Clim, it can be seen that the model residuals gradually shrunk from −4.0 cm and 6.0 cm to −3.0 cm and 2.0 cm, and the model residuals decreased significantly.

### 2.4. Individual Height Growth Model

The conditions and tree species of the study area remained unchanged, and it was convenient and practical to build a machine learning model for a specific research object in the nntool toolbox of MATLAB [54]. Referring to the method of individual diameter growth model construction, the H data set was imported, and the RP, BP, and RF models were directly used to train the H data set to verify the applicability of the model.

#### 2.4.1. Individual Height Growth Model Based on RP

The Gompertz model, Logistic model, Mitscherlich model, and Richards model were selected to fit H models using the least squares method. The R^2^ and RMSE were used to evaluate the fitting results. As shown by the fitting results in Appendix A, the Mitscherlich model had the best fitting effect. The parameter estimations of ω0 and ω1 are shown in Table 4.

The re-parameterized model including Comp, Site, and Clim is as follows.
(5)$$H=\omega_{0}*(1-e^{-\omega_{1}*T})$$

In the formula, ω_0_ and ω_1_ are parameters. The formulae are shown in Equations (2) and (3). The values of the coefficients in the formula are shown in Table 4.

The evaluation index of the H-RP model was calculated, and the R^2^ and RMSE values were 0.683 and 2.102 m. As can be seen from the results, compared to the Mitscherlich model, the R^2^ of the H-RP model improved by 4.1%, and the RMSE decreased by 5.0%. The predicted H changes with T and the residual diagram of the H-RP model are shown in Figure 5, indicating that the H-RP model could reflect the growth process of tree height. From the residual of the model, it can be seen that the residuals of the RP model were uniformly distributed between −3 m and 3 m, and the residual diagram did not have heteroscedasticity.

#### 2.4.2. Individual Height Growth Model Based on BP

The BP model was used to construct an individual tree height model (H-BP model) containing Comp, Site, and Clim, and the predicted H changes in T and the model residual diagram were obtained, as shown in Appendix A. When calculating the evaluation index, the R^2^ was 0.731, RMSE was 1.857 m, and MAE was 1.686 m, which met the accuracy requirements in practice. On the H test set, compared to the RP model, the R^2^ of the BP model improved by 7.0% and the RMSE decreased by 11.7%. From the residual of the model, it can be seen that the residuals of the BP model were uniformly distributed between −4 m and 3 m, and the residual was evenly distributed between −3 m and 3 m; there was no heteroscedasticity.

#### 2.4.3. Individual Height Growth Model Based on RF

The RF model was used to construct an individual tree height model (H-RF) containing Comp, Site, and Clim, and the predicted H changes with T and model residuals were obtained, as shown in Appendix A. When calculating the evaluation index, the R^2^ of the H-RF model for tree height was 0.845, RMSE was 1.267 m, and MAE was 1.153 m. On the H test set, compared to the RP model, the R^2^ of the RF model improved by 23.7%, and the RMSE decreased by 39.7%. These results clearly demonstrate the advantages and potential of machine learning algorithms over traditional models. It can be seen that the residuals of the RF model were uniformly distributed between −2 m and 2 m, model accuracy was significantly improved, and there was no heteroscedasticity in the model.

### 2.5. Performance Evaluations of Three Optimal Models

In order to compare the prediction accuracy of different whole models in individual tree growth modeling, the prediction accuracy of the three models was compared and analyzed (Appendix A).

Figure 6 shows a comparison between the predicted values and the actual values of the three models for DBH and H. It can be seen that the predicted results of the three models were close to the actual values. The BP and RF models were effective for the use of data sets containing a large number of environmental predictors with complex interactions and nonlinear relationships. Among them, the RF model had the best prediction effect with good generalization ability and statistical reliability.

## 3. Discussion

Existing studies on individual tree growth models are constructed using different methods for Chinese fir. Using multiple stepwise regression methods to build the linear mixed-effects model with sample plots as random effects, the R^2^ of the model is 0.676 [55]. Considering climate and competition, an individual tree growth model at DBH was constructed using the Bayesian model with an R^2^ of 0.6121 [56]. The results of our study clearly demonstrated the advantages and potential of machine learning algorithms in the individual growth modeling of Chinese fir. Compared with the formula model, the machine learning model can model the complex nonlinear relationship without being restricted by statistical assumptions. Especially when the model contains independent variables and a large amount of data, it is more convenient to use machine learning in the modeling process and has obvious advantages.

The influence of environmental factors on the growth of an individual tree was analyzed through the results of 10-fold cross-validation. Appendix A shows that, at the level of individual trees, the age of individual trees is the main factor affecting the growth of trees. The relative importance of individual trees was extremely high, exceeding 60%, and the relative importance of competition factors to the model was about 8%.

In Site, PW had the most significant effect. In the BP and RF models, the model accuracy was increased by about 4%. Followed by HB and TRHD, the model accuracy increased by about 3%, slope aspect (PX) and DM had the least effect, and the contribution rate was about 1%. This result was not entirely consistent with those of previous studies [57]. The main reason for this is the difference in the study area. Fujian Province has more hills and mountains, and the vertical variation in heat and precipitation is more pronounced, which determines the growth difference of the study subjects; therefore, PW and HB become important factors.

PW represents the soil erosion and accumulation capacity. Generally speaking, soil temperature gradually decreases from top to bottom, while moisture gradually increases. Chinese fir is an acidic positive tree species that likes deep, fertile, and moist soil and has good drainage conditions, and a down slope position is more conducive. HB represents temperature, elevation increase, and temperature decrease, which is not conducive to the growth of Chinese fir at DBH [58]. TRHD represents soil fertility and water holding capacity. The thicker the soil, the higher the comprehensive fertility and water holding capacity, promoting the growth of DBH. This conclusion is consistent with the findings of Monserud et al. [59]. PX represents light, and light is more sufficient on sunny slope than on shady slopes, and the photosynthesis of Chinese fir is stronger, promoting DBH growth; this conclusion is consistent with the findings of LU [60]. DM represents the soil nutrient distribution and temperature and humidity conditions in the woodland. Compared with hills, Chinese fir has a higher material accumulation capacity in mountains with abundant rainfall and high air humidity [61].

We also found that the partial dependence of a variable on DBH growth is highly correlated with the relative importance of the variable (by comparing the ranking results of characteristic importance). Specifically, when the relative importance of a variable is greater, the average annual diameter growth of individual trees changes more sharply with the change in the variable. When the relative importance of a variable is less, the average annual diameter growth of individual trees changes more smoothly with the change in the variable. At the same time, competition factors, site factors, and climate factors have interactive effects on the growth of individual trees.

## 4. Materials and Methods

### 4.1. Study Area

The study area is Fujian Province, China, and the geographical location map is shown in Figure 7. Fujian Province is located on the southeast coast of China, 115°50′~120°40′ E, 23°33′~28°20′ N. The average annual temperature of the province is 17–21 °C, and it has an annual rainfall of 1400–2000 mm. The altitude of the province is high in the northwest and low in the southeast. Most of the province is hilly, with few plains, and hills and valleys account for more than 80% of the total area. The southeastern coastal region has a south subtropical climate, while the northeast, north, and west parts of the province have a central subtropical climate. The climate characteristics vary greatly between regions in the province. With a forest area of 81,158 km^2^ and a forest coverage rate of 66.80%, Fujian Province ranked first in China for 43 consecutive years. The main tree species include Chinese fir, Masson pine (*Pinus Massoniana Lamb.*), Fokienia hodginsii (*Fokienia Hodginsii (Dunn.) A. Henry & H. H. Thomas*), etc. Among them, Chinese fir forests have the largest area of all tree species. The main soil types mainly include red soil, yellow soil, and lateritic soil, which is suitable for the growth of Chinese fir [62].

### 4.2. Data

#### 4.2.1. Climate Data

The climate factors’ data for the study were generated using the ClimateAP (http://ClimateAP.net, accessed on 22 March 2023) [63] software of Microsoft, providing climate data covering the entire Asia-Pacific region. By entering the longitude, latitude, and elevation annual-scale, climate data for the sample plots of the survey year can be obtained. A total of 17 climate factors were used, and the basic overview is shown in Appendix A.

#### 4.2.2. Sample Plot Data

The sample plot data were collected from the CFI of Fujian Province from 1993 to 2018. According to the “Technical regulations for continuous forest inventory” [64], the sample plots were generally square, with a side length of 25.82 m and an area of 667 m^2^. The measured attributes included individual DBH, H, age, sample plot tree density, and site factors. Site factors include landform, elevation, slope aspect, slope position, slope gradient, soil name, and soil thickness. The summary of sample plot data and individual tree data obtained after data collation is shown in Table 5.

The DBH data set covered 24 plots in different counties from 1993 to 2008. The plots were repeatedly measured four times with an interval of 5 years, and a total of 1432 individual trees were collected. According to the rules of the CFI, H is generally determined by selecting 3–5 trees nearest to the center of the sample plot as a sample for height determination. The study selected 36 plots in different counties; the plots were repeated 3 times with an interval of 5 years, and a total of 357 individual trees were collected. The H data set covered the years from 2008 to 2018.

We obtained the original data set. Due to the old age of some data in the original data set and errors in tree status records, this study followed the Pauta Criterion [65], taking the sample plot as the unit and eliminating non-relevant data according to the triple standard deviation method. Finally, we obtained 1189 DBH data and 307 H data of individual trees. In order to facilitate model construction, DBH data set and H data set were stored as Excel tables in the same format. The DBH data set was taken as an example (Appendix A).

The train_test_split function was used in the scikit-learn machine learning library in Python language to randomly partition the data set [66], with 80% as the training set for model training and 20% as the test set for validating and evaluating the reliability and generalization ability of the model.

### 4.3. Modeling Approaches

#### 4.3.1. Model Independent Variable Selection

The independent variables of the individual tree growth model included four groups of factors: age (T), Comp, Site, and Clim.

According to existing studies [67,68], N is the most important factor affecting the level of competition in a sample plot. N affects the diameter growth of trees, as shown by the decrease in diameter with increasing N. At the same time, N changes light conditions and the soil environment, which also affects the content of soil organic matter. Therefore, N was chosen to quantify competition in the sample plot.

We used RF feature importance evaluation to sort Site and Clim. The basic idea of RF feature importance evaluation is [69]: calculate the reduction value of the Gini coefficient (DGi) for Site and Clim at node splitting, sum the DGi of all nodes in the random forest, then after averaging over all trees to calculate the importance of Site and Clim, and finally normalize and sort according to the importance size. The results as shown in Figure 8.

According to Figure 8, the top 5 factors with the highest relative importance of features in Site were PW, HB, TRHD, PX, and DM. The top 5 factors with the highest relative importance of features in Clim were DD18, MAP, Eref, MCMT, and MAT. 

Secondly, we calculated the Pearson correlation coefficient (Pearson’s r) between each of the Site and the Clim. If r > 0.7, the two factors are strongly correlated; to avoid multicollinearity violating statistical assumptions, only one of the two factors with a higher contribution rate was selected. A thermal map of the correlation coefficient was obtained, as shown in Figure 9.

According to Figure 9, there was no strong correlation between Site variables. In Clim variables, MAT was strongly correlated with Eref, MCMT, and MAP; meanwhile, MCMT was strongly correlated with DD18 and Eref. Therefore, MAT and MCMT were excluded.

After feature importance evaluation ranking and Pearson’s r screening, we discarded unimportant factors and variables with strong autocorrelation. The final Site included PW, HB, TRHD, PX, and DM. The final Clim included DD18, MAP, and Eref.

Therefore, the individual tree growth model driven by competition and environment can be expressed as Equation (6).
(6)y=f(T,Comp,Site,Clim)
where y represents the DBH model or H model, f( ) represents modeling method, T represents age, Comp represents competition factors, Site represents site factors, and Clim represents climate factors.

To analyze the contribution of competition and environment to the dependent variables of DBH and H step-by-step, the following four types of models were, respectively, compared and analyzed:

(1) Sub-model 1, that is, the independent variable only contains T;
(7)y=f(T)

(2) Sub-model 2, that is, the independent variable contains T and Comp;
(8)y=f(T,Comp)

(3) Sub-model 3, that is, the independent variable contains T, Comp and Site;
(9)y=f(T,Comp,Site)

(4) Whole model, that is, the independent variable contains T, Comp, Site, and Clim;
(10)y=f(T,Comp,Site,Clim)

#### 4.3.2. Re-Parameterized Model

In the base model, stand growth is completely age dependent. To quantify the effects of competition, stand, and climate on individual tree growth, an RP method was adopted, in which the parameters (ω0 and ω1) in the base model were expressed as linear combinations of Comp, Site, and Clim. The RP model construction consisted of the following main steps:

Step 1: Base model selection. The Gompertz, Logistic, Mitscherlich, and Richards models were selected as alternative base models. The four base models were fitted using the study data set, and the optimal base model was selected by solving the model parameters using the least squares method. The alternative base models are shown in Table 6.

Step 2. Stepwise regression screening variables. After the basic model was selected, to avoid model multicollinearity, the stepwise regression method was used to remove the largest variable of VIF (Variance Inflation Factor) from the model and then re-fit it. The process was repeated until the VIF < 5 (in general, no collinearity between independent variables can be considered if VIF < 5).

Step 3. Construct the parameterized model. The selected variables were added into the basic model as covariates and fitted through the NLS (Nonlinear Least Squares) function of R software [70] to obtain the form with the best fitting effect.

#### 4.3.3. Back Propagation Neural Network

The essence of a neural network is the approximation of an algorithm or function. It is comprises multiple layers of neurons. Each layer of neurons receives the original input or input from the neuron of the previous layer and then performs mathematical operations and outputs the operation results to the next layer of neurons. The calculation formula of the neuron output value n is shown in Equation (11).
(11)$$n=A(\sum_{i=1}^{t}\omega_{i}m_{i})$$
where t is the input number of neurons in this layer, $\omega_{i}$ is the weight of neuron input in layer i, $m_{i}$ is the input value of neuron in layer i, and A is a predefined nonlinear function.

The BP model was selected in this study, the model was composed of an input layer, output layer, and several hidden layers; each layer contained several neurons, and the neurons between layers were connected by a weight or threshold value. Each layer of neurons affected only the next layer of neurons, and the neurons in the same layer were not connected with each other. The model structure is shown in Figure 10.

The BP model construction process was as follows:

Step 1. Determine the training sample. The data set was divided, with 80% as the training set and 20% as the test set. T, Comp, Site, and Clim were selected as the input variables of the model, and DBH and H were selected as the output variables, respectively.

Step 2. Determine the network structure and parameters. We chose three layers of neural networks, including one input layer, one hidden layer, and one output layer. Equation (12) was used to calculate the number of hidden layer nodes, the three-layer network structure is 10:6:1. Set the transfer function between the input layer and the hidden layer as Log-sigmoid, and set the transfer function between the hidden layer and the output layer as Purelin.
(12)$$l=m+\sqrt{n+o}$$
where n is the number of neurons in the input layer, o is the number of neurons in the output layer, and m is any integer between 1 and 10.

Step 3. Network training. The BP model was trained using the Python language [71]. Select the training data set, call the BP model for training, and compare the influence of different inputs on the fitting accuracy of DBH and H.

In this study, the model results were optimal when the learning rate was set to 0.01, the target accuracy was 0.001, and the maximum number of iterations was 1000.

#### 4.3.4. Random Forest

The RF model is composed of multiple decision trees, and each node in the decision tree is a condition about a feature. The Bootstrap resampling method was used to extract multiple samples from the original data, conduct decision tree modeling for each sample, and combine the prediction of multiple decision trees to jointly predict the results. To compare the difference, the RF model was set up to use the same data set partitioning ratio as the BP model, with 80% as the training set and 20% as the test set, and the mean using the same input variables as the BP model, including T, Comp, Site, and Clim. The RF model was trained using the Python language. The training process of the RF model is as follows.

Step 1. The original data sample content is N. Randomly generate M variables for a binary tree on N nodes. The choice of binary tree variables satisfies the principle of minimum Gini impurity.

Step 2. Use the bootstrap combination method to sample with replacement n sample sets in M to form n decision trees; then, the unsampled samples were used for the prediction of a single decision tree.

Step 3. According to the random forest composed of n decision trees, the final result is the average output of each decision tree.

The RF model has two important parameters to set: the number of decision trees (ntree) and the number of variables randomly selected by tree nodes (mtry). Generally speaking, the overall error rate tended to be stable after ntree reached 500. Meanwhile, Breiman [72] suggested that for regression problems, the default value of mtry should be set to 1/3 of the number of all arguments. Taking this as a reference, we take ntree = 1, 20… 500 to tune ntree. This study used at most 10 independent variables; therefore, 1 ≤ mtry ≤ 10, mtry = 1, 2… 10. The results show that the model works best when ntree = 160 and mtry = 4. 

### 4.4. Model Evaluation

The following fitting statistical indicators were selected as the model selection criteria. The calculation formulas of each test indicator are shown in Equations (13)–(15):

(1) Coefficient of Correlation (R^2^);
(13)$$R^{2}=\frac{\sum_{i=1}^{n}{\left(\hat{y_{i}}-\bar{y}\right)}^{2}}{\sum_{i=1}^{n}{\left(y_{i}-\bar{y}\right)}^{2}}$$

(2) Root Mean Square Error (RMSE);
(14)$$RMSE=\sqrt{\frac{1}{n}*\sum_{i=1}^{n}{\left(y_{i}-\hat{y_{i}}\right)}^{2}}$$

(3) Mean Absolute Error (MAE)
(15)$$MAE=\frac{1}{n}*\sum_{i=1}^{n}\left|\left(y_{i}-\hat{y_{i}}\right)\right|$$
where the $y_{i}$ is the actual value of i, $\hat{y_{i}}$ is the predicted value of i, $\bar{y}$ is the dependent variable of the actual value of the mean, and n is the number of samples.

In addition to the above statistical indicators, we also chose residual plots for the model evaluation. As an important regression diagnostic quantity, the residuals imply important information about the model assumptions. The analysis of residuals can examine the following issues [73]: the feasibility of the linearity assumption of the regression function; the reasonableness of the assumption of equal variance of random errors; the reasonableness of the assumption of independence of errors; the feasibility of the assumption of normal distribution of errors; the presence of outliers in the observations; and the omission of certain important independent variables in data collection or model fitting.

## 5. Conclusions

The re-parameterized model and machine learning model were used to construct the DBH model and H model of Chinese fir. The input variables included T, Comp, Site, and Clim. The results showed that: (1) The addition of competition and environmental factors could improve the prediction accuracy of the individual tree growth model. In terms of site factors, PW had the most significant effect, followed by HB and PX. In terms of climate factors, the DD18 contribution rate was the highest, followed by MAP, and Eref. (2) Among the three models, the RF model fits best. The model constructed in this study could well reflect the growth process of individual DBH and H of Chinese fir in Fujian Province and has reference significance for the growth models of other provinces and other tree species.

Climate factors were added into the individual tree growth model, which fully reflected the spatial differences of climate factors and solved the key problem of applicability of the individual tree growth model in different regions. The model was applied to the forest management decision support system. The results showed that the individual tree growth model, including T, Comp, Site, and Clim, could be used to predict the stand harvest in different sites and climates and provide decision support for harvest prediction and management method selection of mixed uneven-aged forests.

Since the area of Chinese fir plantation forest is the largest in terms of plantation forests in China, this paper only took Chinese fir as an example to construct the individual tree growth model. In future research, the modeling approach in this paper could be applied to the construction of individual tree growth models for other tree species, such as Masson pine (*Pinus massoniana Lamb.*) and broadleaf species. We also recommend using longer study years to investigate the variables to provide more reliable results.

## Figures and Tables

**Figure 1 plants-12-02697-f001:**
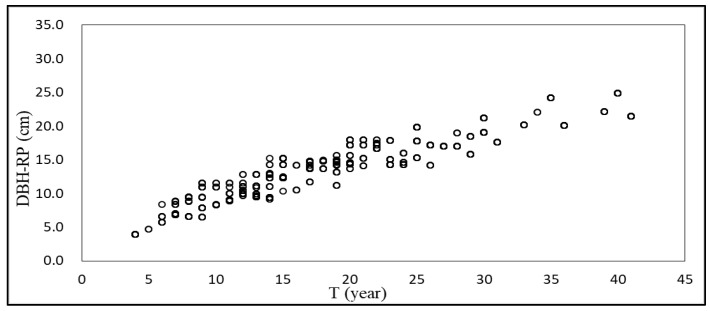
DBH-RP model predicts the change of DBH with T.

**Figure 2 plants-12-02697-f002:**
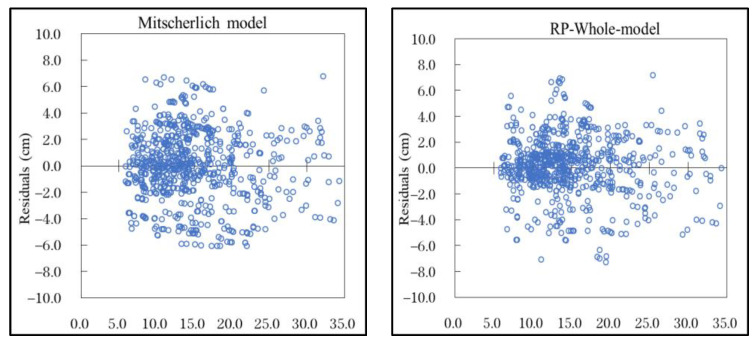
Residual graphs of Mitscherlich model and DBH-RP model.

**Figure 3 plants-12-02697-f003:**
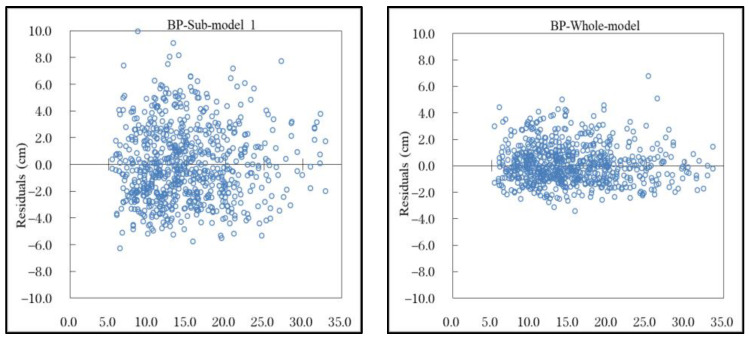
Residual diagram of sub-model 1 and whole model based on BP model. Residual diagram of sub-model 2 and sub-model 3 are presented in Appendix A.

**Figure 4 plants-12-02697-f004:**
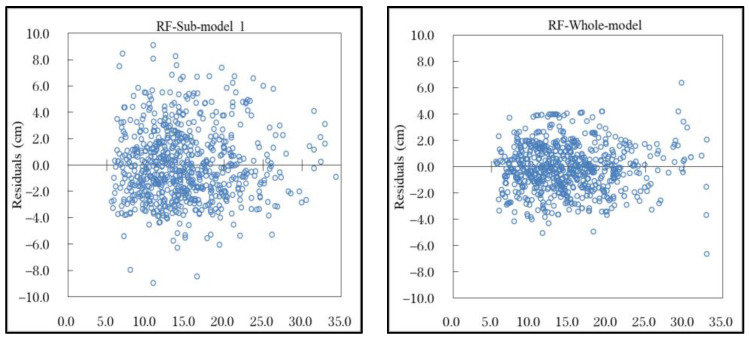
Residual diagram of sub-model 1 and whole model based on RF model. The residual diagrams of sub-model 2 and sub-model 3 are presented in Appendix A.

**Figure 5 plants-12-02697-f005:**
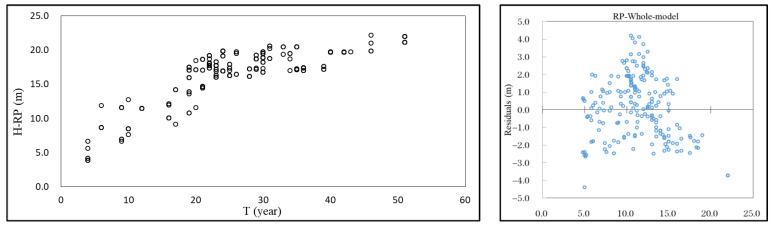
Tree height variation with age predicted by RP model and residual of model.

**Figure 6 plants-12-02697-f006:**
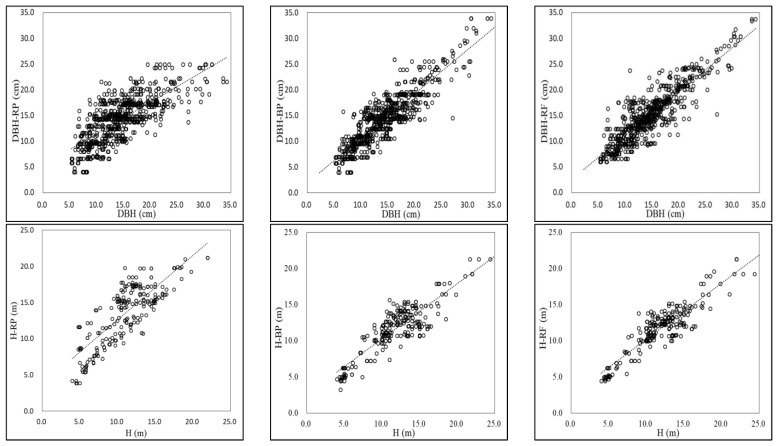
Comparative analysis of predicted and actual values of DBH and H by three models.

**Figure 7 plants-12-02697-f007:**
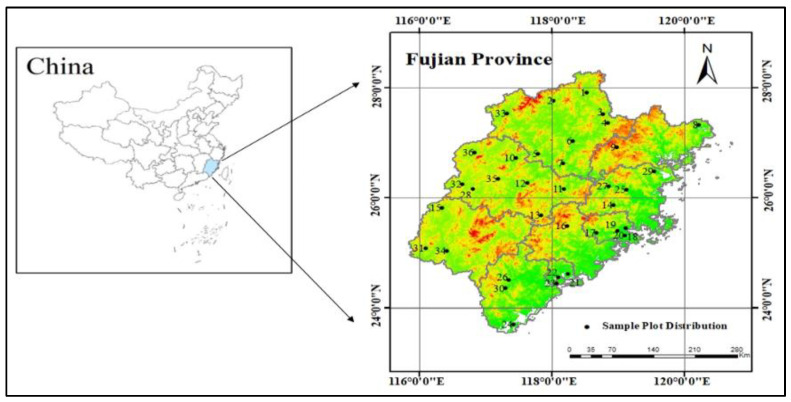
Geographical map of the study area and sample plot distribution in the Fujian Province of China. Among them, numbers 1–24 represent the location distribution of sample plots in the DBH data set, and numbers 1–36 represent the location distribution of sample plots in the H data set.

**Figure 8 plants-12-02697-f008:**
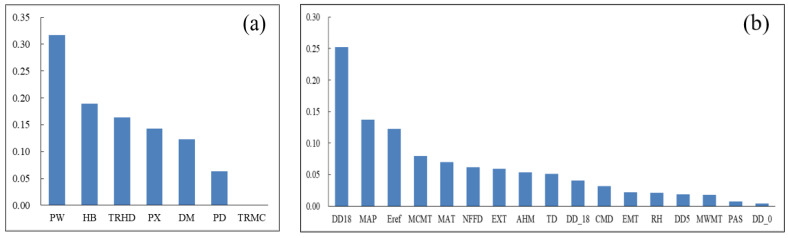
Rank of importance of environment variables. Where (**a**) ranks the importance of Site, and (**b**) ranks the characteristic importance of Clim.

**Figure 9 plants-12-02697-f009:**
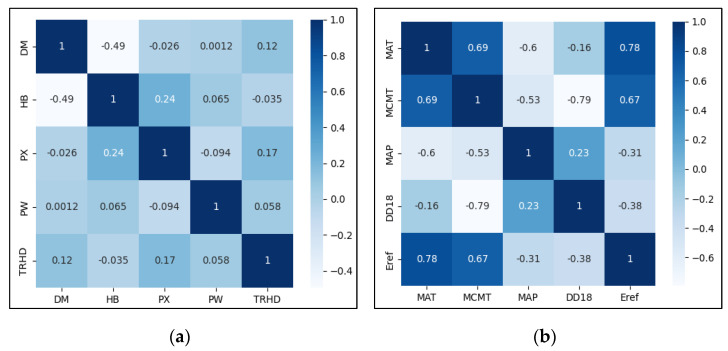
Thermal maps of environmental variables with correlation coefficients. Where (**a**) represents thermal maps of Site variables with correlation coefficients, and (**b**) represents thermal maps of Clim variables with correlation coefficients.

**Figure 10 plants-12-02697-f010:**
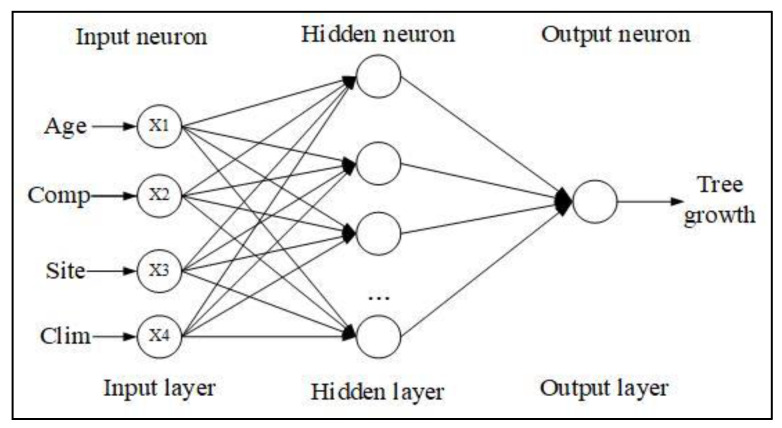
Structure diagram of status level exponential model of 3-layer BP model.

**Table 1 plants-12-02697-t001:** Parameter estimation of the DBH-RP model.

Parameter	Intercept	Covariates						
ω0	α0	α1	α2	α3	α4	α5	α6	α7
	−20.041	0.001	−0.096	−0.004	0.155	−1.525	0.001	0.031
ω1	β0	β1	β2	β3	β4	β5	β6	β7
	0.039	5.89 × 10^-6^	−4.11 × 10^-5^	4.01 × 10^-6^	−6.17 × 10^-6^	0.000	4.79 × 10^-6^	−7.63 × 10^-6^

**Table 2 plants-12-02697-t002:** Fitting results of the individual diameter growth model based on BP.

Model	Type of Variable	Train Evaluation			Test Evaluation		
		R^2^	MAE (cm)	RMSE (cm)	R^2^	MAE (cm)	RMSE (cm)
Sub-model 1	T	0.669	2.041	2.626	0.638	2.164	2.745
Sub-model 2	T + Comp	0.721	1.836	2.375	0.689	1.962	2.506
Sub-model 3	T + Comp + Site	0.809	1.463	1.973	0.776	1.614	2.141
Whole model	T + Comp + Site + Clim	0.831	1.342	1.824	0.808	1.487	2.017

**Table 3 plants-12-02697-t003:** Fitting results of individual diameter growth model based on RF.

Model	Type of Variable	Train Evaluation			Test Evaluation		
		R^2^	MAE (cm)	RMSE (cm)	R^2^	MAE (cm)	RMSE (cm)
Sub-model 1	T	0.714	1.893	2.391	0.679	1.997	2.537
Sub-model 2	T + Comp	0.782	1.593	2.104	0.733	1.782	2.281
Sub-model 3	T + Comp + Site	0.827	1.368	1.866	0.826	1.387	1.871
Whole model	T + Comp + Site + Clim	0.867	1.201	1.573	0.849	1.267	1.691

**Table 4 plants-12-02697-t004:** Parameter estimation of the H-RP model.

Parameter	Intercepts	Covariates						
ω0	α0	α1	α2	α3	α4	α5	α6	α7
	−2.301	0.014	0.625	−0.002	0.095	1.737	−0.011	0.012
ω1	β0	β1	β2	β3	β4	β5	β6	β7
	0.063	9.62 × 10^-7^	−6.83 × 10^-5^	−1.62 × 10^-6^	4.68 × 10^-5^	0.000	−1.01 × 10^-5^	2.63 × 10^-5^

**Table 5 plants-12-02697-t005:** Site factors data and individual tree data statistics in DBH data set and H data set.

DBH Data Set					H Data Set				
Variable	Description	Statistic ^2^			Variable	Description	Statistic ^2^		
		Max	Min	Mean			Max	Min	Mean
T	Age	41	4	23	T	Age	51	4	25
DBH	Diameter at breast height	35.4 cm	5.3 cm	15.6 cm	H	Height	22.0 m	3.5 m	9.7 m
N	Sample plot tree density	238	40	121	N	Sample plot tree density	282	25	117
HB	Elevation	885 m	100 m	423 m	HB	Elevation	1041 m	101 m	730 m
PD	Slope gradient	35°	2°	9°	PD	Slope gradient	37°	5°	25°
TRHD	Soil thickness	120 mm	73 mm	98 mm	TRHD	Soil thickness	130 mm	60 mm	101 mm
DM ^1^	Landform	Middle Mountains (3), Low mountains (4), Hills (5)							
PX ^1^	Slope aspect	North (1), Northeast (2), East (3), Southeast (4), South (5), Southwest (6), West (7), Northwest (8)							
PW ^1^	Slope position	Ridge (1), Upper (2), Middle (3), Lower (4), Valley (5), Flat (6)							
TRMC ^1^	Soil name	Red soil (12), Lateritic soil (13)							

^1^ In order to facilitate data storage and calculation, refer to the investigation factor code of fixed sample plots in the Operational Rules for Review of National Forest Resources Continuous Inventory in Fujian Province. ( ) stores the codes of the non-quantified site factors, including DM, PX, PW, and TRMC in the data set. ^2^ Max stands for maximum value; Min stands for minimum value; Mean stands for Average value.

**Table 6 plants-12-02697-t006:** Alternative basic models.

ID	Model	Equation ^1^
1	Gompertz	$y=a*e^{-b*e^{-c*t}}$
2	Logistic	$y=a/(1+b*e^{-c*t})$
3	Mitscherlich	$y=a*(1-e^{-b*t})$
4	Richards	$y=a*{\left(1-e^{-c*t}\right)}^{b}$

^1^ In the equation, y represents the independent variable, t represents the age, and a, b, and c represent the parameter.

## Data Availability

The data is not publicly available because it relates to the copyright of the data provider.

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
