# Peer review of "Regionally Compatible Individual Tree Growth Model under the Combined Influence of Environment and Competition"

_plants, 2023, doi:10.3390/plants12142697_

Round 1

Reviewer 1 Report

In the conditions of dynamically changing natural and economic factors, the development of tree and stand growth models is an important direction of research in the field of forest resource management. They not only reduce labor-intensive direct measurements of trees, but also allow to provide reliable information needed for planning the development of wood resources in plantation forests. Therefore, the article presented for review concerns an important issue of practical assessment and forecasting of tree growth based on the analysis of environmental factors.

Therefore, the efforts of the authors of the article, who presented various methods of constructing tree growth models, taking into account many environmental factors, should be appreciated. This allowed both the comparison of model development technologies and the analysis of model prediction accuracy. For this purpose, the chapter "Materials and methods" describes the factors used and methodological assumptions quite well. Despite this, in the methodological part, the authors did not fully explain the adopted solutions. For example, they did not write

• How big were the sample plots? Was the tree density determined for the sample plot, or was the average stand density used for the calculations (Table 2)

• Were selection fellings of trees on the forest plantation carried out during the research period? The felling of trees has an impact on density and thus on competitiveness

The obtained results, due to the short period of research, do not provide grounds for creating general models. Reliable results are provided only by long-term observations covering the entire tree development cycle. It should be emphasized in the article that these are preliminary research results, because some of the analyzed factors are characterized by variability that can be observed over a longer period of time (e.g. incremental reaction of trees at different densities, climate change)

Overall, however, the results were well presented and the whole article is very well done

Reviewer 2 Report

Article: Regionally compatible individual tree growth model under the combined influence of environment and competition

General comments:

The article deals with the calculation of a individual tree model combined with the influence of environmental variables and competition among trees.

The results obtained are good and consistent with this type of study on forest species. However, the authors indicate that other similar studies already exist (although few).

The article is well explained in detail the methodology followed, although it presents some aspects that discourage its publication in Plants journal:

- It is not novel enough and its quality is not enough to be published in a high-impact scientific journal.

- The number of plots where the DBH were measured is low (16 plots).

- It presents defects in the structure of its content such as some results (page 17) that appear in the discussion.

- It presents many repetitive figures that are not very relevant.

It is recommended that these aspects be reviewed and sent it to another more appropriate journal.

Specific comments:

-          Page 2, line 50: to correct “… machines(SVM)…” to “… machines (SVM)…”

-          Page 2, line 51: to correct “… network(NN)…” to “… network (NN)…”

-          Page 3, line 120: to corret “… CunnInghamia Lancelata…” to “…Cunninghamia Lanceolata…”

-          Table 4, Table 5 and Table 9: to correct “Compertz” to “Gompertz”

-          Please, revise the format of the List of References

Reviewer 3 Report

The manuscript entitled “Regionally compatible individual tree growth model under the combined influence of environment and competition” reflects the development of applied research on the topic of forest tree modelling. However, introduction, materials and methods, results and discussion need to be improved. Thus, major changes are recommended.

Comments

1) Line 36 – mingled forests or mixed forests?

2) Lines 40-42 – please clarify text.

3) Line 111 – what do the authors mean by “intermittent cutting“? Periodical cuttings?

4) Lines 127-141 – please state clearly the objectives.

5) Lines 149-150 – please clarify text.

6) Lines 160-165 – please clarify text.

7) Lines 194-197 – please clarify text. Also, the reference of the software should be included.

8) Lines 202-206 – please clarify text.

9) Lines 207-209 and Figure 2 – the text needs further details. The importance of the environmental variables is ranked as function of what?

10) Lines 226-229 – please clarify text.

11) Lines 243-253 – please clarify text.

12) All the software’s used should have a references (e.g., R software, SPSS)

13) Lines 265-266 – describe with further details.

14) Lines 283-296 – please clarify text.

15) Lines 297-318 – please clarify text. It is not clear whether the authors have split the data set in training and fitting and the parameters used for fitting the model.

16) Software used to fit Random Forest and Neural Network are missing.

17) In the Results section there are methods that should in in the methods section. For example, residual graphics, text of lines 371-374, 400-406.

18) Lines 34-350 – please clarify text.

19) Line 354 has the same information as table 6.

20) Lines 364-367 – the text does not seem to correspond to figure 6.

21) Line 382 – what do the authors mean by “certain biological significance”? Please clarify text.

22) Lines 416-423 – please clarify text.

23) Lines 455-468 – compared to the results of the diameter modelling those of the height modelling could be improved with a more detailed description.

24) Discussion section is a summary of the results. In this section the results of this study should be discussed with published references.

25) Lines 558-561 – please clarify text.

Round 2

Reviewer 2 Report

The authors have made a great effort to improve the article, especially redoing all the calculations by including 8 new plots, thus correcting one of the biggest problems the article had, and rewriting many parts of it.

However, the article continues to have major flaws such as an excessive number of repetitive and insignificant figures and, above all, the discussion in which some results (figures) are included and which is not very consistent for a high-impact article.

Therefore, I recommend authors to submit the article to another more appropriate journal.

Reviewer 3 Report

The second version of the manuscript improvements were made and most of the comments were answered. Yet, there are still some issues that should be improved. Thus, minor changes were recommended.

 Comments

1) Lines 170-171 – “The topography of the province is high in the northwest and low in the southeast.” Are the authors refereeing to altitude?

2) Software used to fit Random Forest and Neural Network should be in methods section.
